# “Reading Pictures Instead of Looking”: RGB-D Image-Based Action Recognition via Capsule Network and Kalman Filter

**DOI:** 10.3390/s21062217

**Published:** 2021-03-22

**Authors:** Botong Zhao, Yanjie Wang, Keke Su, Hong Ren, Haichao Sun

**Affiliations:** 1Changchun Institute of Optics, Fine Mechanics and Physics, Chinese Academy of Sciences, Changchun 130033, China; zhaobotong19@mails.ucas.ac.cn (B.Z.); sukeke19@mails.ucas.ac.cn (K.S.); renv587@126.com (H.R.); ciomp_shc@163.com (H.S.); 2University of Chinese Academy of Sciences, Beijing 100049, China

**Keywords:** human posture estimation, capsule network, 6D object pose estimation, Kalman filter

## Abstract

This paper proposes an action recognition algorithm based on the capsule network and Kalman filter called “Reading Pictures Instead of Looking” (RPIL). This method resolves the convolutional neural network’s over sensitivity to rotation and scaling and increases the interpretability of the model as per the spatial coordinates in graphics. The capsule network is first used to obtain the components of the target human body. The detected parts and their attribute parameters (e.g., spatial coordinates, color) are then analyzed by Bert. A Kalman filter analyzes the predicted capsules and filters out any misinformation to prevent the action recognition results from being affected by incorrectly predicted capsules. The parameters between neuron layers are evaluated, then the structure is pruned into a dendritic network to enhance the computational efficiency of the algorithm. This minimizes the dependence of in-depth learning on the random features extracted by the CNN without sacrificing the model’s accuracy. The association between hidden layers of the neural network is also explained. With a 90% observation rate, the OAD dataset test precision is 83.3%, the ChaLearn Gesture dataset test precision is 72.2%, and the G3D dataset test precision is 86.5%. The RPILNet also satisfies real-time operation requirements (>30 fps).

## 1. Introduction

Machine vision has been widely used in the imaging field, particularly as artificial intelligence grows increasingly advanced. Action recognition, a notable research direction related to machine vision, includes the detection, recognition, and analysis of targets in image sequences. It is widely used in human–computer interactions, such as medical rehabilitation, dancing instruction, and traffic supervision. Its efficacy is impacted, however, by issues with target diversity and background complexity.

There are two main methods of traditional action recognition. The first is based on template matching for geometric calculation, which is operated mainly by artificial and high-dimensional modeling for comparison among various parts of the human body in static images to find their spatial correspondence. The second is graph structure based; these techniques have relatively low complexity but are sensitive to noise and other factors which drive down their accuracy. In 2012, Alex’s convolutional neural network (CNN) dominated the ImageNet competition [1], proving the value of deep learning in the image processing field. The traditional template matching method has been largely replaced by deep learning methods.

In 2013, Toshev’s team proposed DeepPose [2] as a cascaded CNN for action recognition. In recent years, advancements in deep learning have created two main research directions in regard to action recognition: bottom to top, where body parts are detected before the posture (e.g., the OpenPose [3] project at Carnegie Mellon University), and top to bottom, where the human body is first positioned and then split into parts to analyze its posture. CNNs reflect the parallelism of features, that is, when a feature is valid in one place, it should be effective in other places. However, they are sensitive to the rotation and scaling of the target. They are also very dependent on random features and require a large number of operations to complete the detection process. There is still a significant difference between CNN results and human observation results.

To mitigate the shortcomings of the CNN, the three-dimensional (3D) space can be converted into a six- or higher-dimensional space. Alternately, the training set can be given target data for rotation and scaling. Unfortunately, both of these approaches increase the calculation burden. In this study, we attempted to improve the CNN for action recognition based on a combination of a capsule network and Kalman filter. The capsule network focuses on the spatial relationship between the target and its parts, where each capsule represents the target part and its parameters. The spatial posture is introduced to remedy the CNN’s over-sensitivity to changes in the characteristic posture and noise. Additionally, capsule networks can build parse trees similar to CNNs. After the capsule network is trained, the results are compared with a Kalman filter model based on previously obtained limb information. If the results are similar, the capsule network information is saved and supplied to a new Kalman filter to check the coordinate points. If the predictions diverge from the Kalman filter, they are deleted or marked as a singular point to improve the accuracy. The Kalman filter is designed to avoid any location points that are too far offset to prevent them from influencing the posture predictions.

The main contributions of this work can be summarized as follows. First, the natural language processing (NLP) concept is used to obtain and analyze global image features rather than relying on a simple convolution. The capsule network enhances the interpretability and adjustability of the model while resolving the sensitivity of the traditional CNN to angle changes and noise. An evaluation model is established with a tree-like network structure, which gives it a swift running speed and convenient analysis of the relationship between layers. Finally, a Kalman filter effectively exploits the features of video sequences to prevent individual errors from affecting the final predictions.

## 2. Related Works

### 2.1. Human Posture Estimation

Action recognition is an important part of many human–computer interactions and patient care applications [4,5,6,7]. Action recognition includes coordinate regression, thermography detection, and hybrid regression and detection models. Models based on coordinate regression include the multi-stage direct regression represented by DeepPose [2], and multi-stage distributed regressions such as IEF [8]. There are several types of action recognition models based on thermal image detection: graph structure models [9], tree structure models [10], implicit-learning structures based on sequential convolution [11], and hourglass network structures [12]. Regression and detection hybrid models may be structured in series [13] or in parallel [14].

### 2.2. 6D Object Pose Estimation

The method of 6D attitude detection is widely used in target recognition [15,16,17,18,19], robot control [20,21,22], autonomous navigation [23,24,25], and augmented reality [26,27]. The 6D pose detection methods may be based on RGB images, depth map/point clouds, or RGB-D data. Traditional pose detection methods based on RGB images mainly rely on the detection of key points and learned features [22,28,29,30,31]. Two-dimensional key points can be predicted by learning [32,33,34,35,36] and poses can be solved by PNP [37].

Recent studies have centered on pose detection based on 3D objects [36,38], where information is obtained through 3D space. For example, PointNets [39] and VoxelNet [40] use a PointNet-like [41] structure to analyze point cloud data and obtain the 6D poses of targets. Other classical 3D feature extraction methods rely on RGB-D data [42,43,44,45]. New methods include PointFusion [25], 6D pack [46], DenseFusion [47], and PoseCNN [48]. We refer to the DenseFusion methodology in this study, which achieves fusion by using the features extracted from an RBG image and depth image.

### 2.3. DenseFusion

Li Feifei’s DenseFusion [47] is a 6D object pose estimation algorithm wherein a dense pixel level fusion process integrates the features of RGB and point cloud data. According to the characteristics of the RGB image and point cloud, which belong to different feature spaces, two data processing techniques are deployed separately. The structure of the two sets of data is retained as are their respective discriminative features after the fusion is complete.

### 2.4. Capsule Network

The capsule network [49] indicates whether an object exists or not, then learns which entity it should represent and some of its parameters. This creates abundant structures in the neural network to enhance the model’s generalization. A capsule consists of three parts: a logistic unit indicates whether the current picture has a target (as the capsule can work anywhere in the image, it can be comprised of CNN modules), a matrix representing the pose of the target, and a vector representing other attributes (e.g., deformation, speed, color). The capsule network does not have the shortcomings of deep learning, which is sensitive to changes in feature angles and noise; it exploits the concept of spatial coordinates in computer graphics. Each layer is connected by a transformer [50], which resolves the problem that the RNN model cannot be parallelized. The results are output after analyzing all the inputs and the correlation features between them.

## 3. Proposed Methods

A regular position matrix is added to the RGB image to mitigate the self-attention’s insensitivity to the position order. This allows each pixel of the image to be introduced into the conceptual position. After obtaining the discriminative features by DenseFusion, the constellation capsule network is obtained by the transformer, including 25 joint points and their coordinates and rotation information. The low-level capsules provide the high-level capsules. The process is shown in Figure 1.

On the other hand, the action is a continuous sequence, so a Kalman filter can be used to predict the position range of the next moment while protecting the final result from any detection error. After obtaining the action information of all frames, a position matrix is added to each frame information and the final result is obtained via transformer.

We use the word “reading” here because we assert that the image processing technique does not involve simply regarding every pixel of the image, but rather, analyzing the information contained in different locations in the image—similar to reading an article rather than looking at a sheet of paper with the article printed on it. We believe this process is similar to operating the sequence model, so we chose a transformer to analyze the capsule network. The transformer enhances the model’s accuracy, but also makes the model more complex. Thus, the model’s computational speed is slower than that of a model using a fully connected layer. The pruning process (Section 3.3) improves the speed of the model without sacrificing its accuracy.

### 3.1. Transformer

The transformer proposed by Ashish Vaswani et al. [50] was utilized in this study. This transformer uses the CNN concept because the LSTM and other classical RNN modules are not easy to parallelize. The module principle is shown in Figure 2.

Each time the model extracts the features of the current vector, it needs to consider the features of other vectors in the sequence to obtain new features. To properly regard the input sequence of each vector as a feature, the model uses a different position matrix for each feature vector. This process can be expressed as follows:(1)A=SIREN(X+Station),
$$Q=W_{q}A+b_{q},$$
$$K=W_{k}A+b_{k},$$
$$V=W_{v}A+b_{v},$$
(2)$$\mathrm{Output}=\mathrm{softmax}(\frac{QK^{T}}{\sqrt{d_{k}}})V,$$

In Formula (1), A represents the activation of input X after adding the position weight matrix Station. (The activation function SIREN is introduced in Section 3.2. The transformer function maps a query and a set of key-value pairs to an output, where the query, keys, values, and output are all vectors.

In the transformer, the matrix of Q contains a set of queries packed together. Keys and values are also packed together into matrixes K and V. *d_k_* denotes the dimension of keys. 

The advantage of transformer module is that it can process features in parallel. As with the CNN, the same knowledge should be used at all image locations. The transformer also uses the Seq2seq [51] concept to ensure that the new features of each output are obtained after summarizing and analyzing the global features.

### 3.2. SIREN Activation Function

The activation function plays an important role in the CNN as it eliminates the linear model while introducing neurons into nonlinear functions. Different activation functions have different effects on network training. In recent years, problems with the Sigmoid function (e.g., function complexity, parameter interdependence, overfitting) have made the ReLU increasingly popular. However, the ReLU function cannot be used to model continuously differentiable signals. The ReLU function cannot learn high-frequency features, though this type of signal has obvious advantages in neural network training.

In this study, we used the periodic nonlinear function SIREN [52] as the activation function to represent the neurons in the hidden layer of the MLP:(3)$$SIREN(x)=W_{n}(\phi_{n-1}\circ \phi_{n-2}\circ \ldots \circ \phi_{0})(x)+b_{n},$$
$$x_{i}\to \phi_{i}(x_{i})=\sin(W_{i}x_{i}+b_{i}),$$
where $\phi_{i}$ represents the *i*th nerve layer.

Compared with Sigmoid, Tanh, and ReLU activation functions, the SIREN function can process natural signals better and converge more stably. The derivative of SIREN is still the SIREN itself, so it has the same characteristics.

### 3.3. Action Recognition Based on Capsule Network

The advantage of CNN is its modularization and parallelism of features. However, the CNN overemphasizes the invariance of features, which makes it sensitive to rotation, scaling, and noise. Unlike graphics, the line rotation or scaling can still be detected (e.g., via Hough Transform [53]). The CNN does not analyze images, per se; it is more likely to rely on a large number of random convolution operations to obtain effective learning features. The interpretability of this process is relatively poor.

For example, as shown in Figure 3, the angle of observation will affect the result of the judgement. At the same time, as shown in Figure 4, the combination of parts will also affect the judgement result.

Considering the above shortcomings, CNNs are generally designed with large training sets or combined with more complex models to obtain more random features. However, this requires additional calculations and may lead to uncertainties or overfitting. The CNN ignores the viewpoint equivariant between the observer and the target. For example, by observing changes in angle, a square can be perceived as a diamond with an angle of 90° or a rectangle with the same length and width. The CNN also ignores the viewpoint invariance between the target and the target part, such as the relative positions of facial features and limbs relative to the human body.
$$\mathrm{Equivariant}:\;\forall_{T\in \Gamma }Tf(x)=f(Tx),\;\mathrm{Invariant}:\;\forall_{T\in \Gamma }Tf(x)=f(Tx),$$

The CNN behaves quite differently than human visual analysis. In the analysis of human body posture, the relative posture between joints, spatial relationships, and observation angles can critically alter the final result. To effectively utilize the information contained in the joints and trunk of the human body, images were analyzed in this study according to the NLP to reduce the dependence on the random features of the CNN. The model regards the pose and attribute of each joint point as a word and the image as a sequence. The information of joint points is used as an input; the body information of the current frame is obtained via Bert. Using low-level capsules to derive high-level capsules avoids the dependence of the CNN on random features and enhances the interpretability of the model.

After obtaining the feature map by CNN, PointNet, and MLP, the transformer determines whether the capsules in the first layer are activated with a permutation-invariant encoder *h^caps^*. The first layer of the capsule includes the coordinates of joint points and related attributes. 

An object capsule has three parts: a 3 × 3 matrix *OV* that represents the relationship between the object and viewer, the capsule’s feature vector *c_k_*, and the probability *a_k_* of its presence.

The capsule parameters are predicted as follows:(4)$$OV_{1:K},c_{1:K},a_{1:K}=h^{caps}(x_{1:M}),$$

Additionally, the candidate parameters from *c_k_* are decoded as:(5)$$OP_{k,1:N},a_{k,1:N},\lambda_{k,1:N}=h_{k}^{part}(c_{k}),$$

A part pose candidate is then decoded as:(6)$$V_{k,n}=OV_{k}OP_{k,n},$$

Additionally, the candidates are turned into mixed components by:(7)$$p(x_{m}|k,n)=N(x_{m}|\mu_{k,n},\lambda_{k,1:N}),$$
(8)$$p(x_{1:M})=\prod_{m=1}^{M}\sum_{k=1}^{K}\sum_{n=1}^{N}\frac{a_{k}a_{k,n}}{\sum_{i}a_{i}\sum_{i,j}a_{i,j}}p(x_{m}|k,n),$$

Every object capsule uses a separate multilayer perceptron (MLP) $h_{k}^{part}$ to predict part candidates from the capsule feature vector c_k_. A candidate capsule includes the probability of capsule activation *a_k_*, a 3 × 3 matrix representing the relationship matrix *OP* between the object and the part, and an associated scalar standard deviation *λ_k_*_,*n*_. $\mu_{k,n}$ denotes candidate predictions, which are given by the product of the matrixes *OV* and *OP*. The training process is likelihood based. The model uses unsupervised learning to select the closest target among the trained targets, then uses the trained target to reconstruct and interpret the high-level capsule. 

The part capsule is formed by a six-dimensional pose *x* (two rotations, two translations, scale and shear), a presence variable *d* ∈ [0, 1], and a set *z* that represents the capsule’s features.

The part capsule’s parameters are predicted as:(9)$$x_{1:M},d_{1:M},z_{1:M}=h^{enc}(y),$$

The color of the *m*th template is predicted as:(10)$$c_{m}=MLP(z_{m}),$$

Affine transforms are applied to the image templates:(11)$$T_{m}=Transform(x_{m}),$$

Mixing probabilities are computed as:(12)$$p_{m,i,j}^{y}\propto d_{m}T_{m,i,j}^{a},$$

Additionally, then the image likelihood is calculated as:(13)$$p(y)=\prod_{i,j}\sum_{m=1}^{M}p_{m,i,j}^{y}N(y_{i,j}|c_{m}\cdot T_{m,i,j}^{c};\sigma_{y}^{2}),$$

After training, the parameters of the network are evaluated and unnecessary parameters are deleted. This allows the model to work at a rapid speed. For example, posture analysis of the forearm only requires observation of the elbow and wrist joints, while the surrender action only requires focus on the upper body or arms. The whole network can be optimized by regularization and batch standardization, but the structure does not need to be adjusted.

Figure 5 shows the posture of two actions at a certain frame.

The score *S* is evaluated as per its contribution to the high-level capsule, not for a certain neuron:(14)$$S_{i,j}=\frac{1}{Ma}\sum_{m=1}^{M}\sum_{n=1}^{a}\frac{Z_{i,j,n}(b_{i})}{Z_{j,n}},$$
$$\sum_{i=1}^{k}S_{i,j}>0.85,$$
where *Z_i,j,n_*(*b_i_*) is the contribution of the combination of the *i*th low-level capsule to the *n*th feature of the *j*th high-level capsule. All features are scored by this method. The average score is taken as the contribution of current low-level capsules to current advanced capsules; the capsules are sorted according to the score. When the top low-level capsules sum more than 85%, the remaining capsules are discarded. The remaining low-level capsules are then used as the basis for judging the activation of the current high-level capsule. As shown in Figure 6.

The evaluation model allows for clarification of the relationship between high-level capsules and low-level capsules, which minimizes the influence of unnecessary low-level capsules on the process of combining high-level capsules. After retraining, the network realizes that, when some low-level capsules are activated and meet certain conditions, high-level capsules can be combined and activated to determine their characteristics.

Experimental results show that the performance of the tree network structure satisfies practical application requirements with relatively quick calculation speed.

### 3.4. Kalman Filter

Action recognition can be carried out on a single frame image, but the features of the sequence are not used. In a video, the target’s action is continuous. We believe the use of Kalman filtering can constrain the predicted coordinates. It prevents any erroneous prediction at a certain moment from creating uncertainty in the result of the entire sequence.

In this study, we used a median filter to denoise the coordinate information of the first 30 frames, then used the filtered result as the initial point to build a Kalman filter to predict the motion range of the joint point at the next moment. The prediction results of the Kalman filter were used as the reference range; the predicted coordinates within the range could then be regarded as the correct points. Any incorrect prediction of the coordinates of the joint points in the current frame were used to check the detection results. As shown in Figure 7.

The state equation of Kalman filter system is as follows:(15)$$x_{k}=Ax_{k-1}+Bu_{k-1}+w_{k-1},$$
which infers the current state according to the state and control variables of the previous time. *x_k_* and *x_k_*_−1_ denote the object’s coordinate of the current and last moment, respectively. *u* is the optional control input, which is generally ignored in actual use. *w_k_*_−1_ is Gaussian noise that is generated in the prediction process.

The observation equation is as follows:(16)$$z_{k}=Hx_{k}+v_{k},$$
where *v_k_* is the observation noise obeying a Gaussian distribution. The time update equation of the Kalman filter is:(17)$$x\prime_{\bar{k}}=Ax\prime_{k-1}+Bu_{k-1},$$
(18)$$P_{\bar{k}}=AP_{k-1}A^{T}+Q,$$
and the state of the Kalman filter is updated as:(19)$$K_{k}=\frac{P_{\bar{k}}H^{T}}{HP_{\bar{k}}H^{T}+R},$$
(20)$$x\prime_{k}=x\prime_{\bar{k}}+K_{k}(z_{k}-Hx\prime_{\bar{k}}),$$
(21)$$P_{k}=(I-K_{k}H)P_{\bar{k}},$$

In our model, a Kalman filter checks the predicted points and curve fitting shows the movement trajectory of a part in the video. The part capsule already contains the position information of the human body, so it does not predict the position of the human object in the image. However, we can use a constellation capsule to obtain the location of the body and the size of positioning frame through transfer learning.

Finally, the capsules processed via Kalman filter are taken as an input and analyzed by the transformer to deduce the posture represented by the video.

## 4. Experiments

We tested the proposed method on the OAD dataset [54], ChaLearn Gesture dataset [55], and G3D dataset [56]. We also compared it against other models with different observation rates.

(1)SSNet [57], a network based on skeleton motion prediction. The network uses start point regression results to select the appropriate layer at each time step to cover the execution part of the action currently in progress. Multi-layer structured skeleton representations are used in the network.(2)FSNet [57], which uses the top layer to predict the action directly; the prediction is based on the fixed window scale. To ensure objectivity, different sizes of windows were used for performance comparison (S = 15, 21, 63, 127, 255).(3)FSNet-MultiNet [57], an upgraded version of FSNet that uses different sizes of scales for repeated detection for enhanced accuracy.All of these architectures use multi-layer structured skeletons for comparison. In order to make a more objective comparison, we used a similar concept for further comparative experiments.(4)ST-LSTM [4], which has shown excellent performance in motion recognition based on 3D skeletons. We adjusted it for an action-prediction task in accordance with our experimental conditions.(5)JCR-RNN [5], a variant of LSTM which models context dependence in the temporal dimension of an untrimmed sequence. It has shown remarkable performance on skeleton sequences of certain benchmark datasets.(6)Attention Net [6], where an attention mechanism dynamically assigns weights to different frames and joints to classify actions based on 3D skeletons. It produces a prediction of the type of action at every moment. However, the interpretability between the low-level network and the high-level network is not realized and the network structure is not pruned.

The prediction accuracy of the observation ratio p%, as reported here, represents the average prediction accuracy in the observation interval of the action instance. As shown in Figure 8.

### 4.1. Experimental Comparison on OAD Dataset

The OAD dataset was collected using Kinect V2 in a quotidian living environment. Ten movement training sessions were conducted by different subjects. The long video sequences in this dataset correspond to about 700 action instances. The start and end frames of each action are marked in the dataset. Thirty long sequences were used for training and 20 long sequences for testing. The prediction results of different models on the OAD dataset are shown in Table 1.

The proposed RPILNet produced the best prediction results under different observation rates. When the observation rate was only 10%, RPILNet performed with accuracy of 68%, which is better than SSNet or FSNet. It also appears to be superior to JCR-RNN or ST-LSTM based on RNN/LSTM, which can handle continuous skeleton sequences. The performance differences between models can be explained as follows.

In the early stages (for example, when observation ratio is 10%), RPILNet focuses on the executed part of the current action. The JCR-RNN and ST-LSTM in the RNN model may bring information about previous actions at this point, which interferes with the current operations. The accuracy of RPILNet can be further improved by Kalman filtering to prevent any individual errors from affecting the final results. In the later stages (for example, when the observation ratio is 90%), the information learned in the early stages of the current action may gradually disappear in the RNN model. RPILNet, however, uses the transformer to analyze all input vectors at the same time and make them refer to each other. This retains knowledge and ensures that all outputs are based on global characteristics.

### 4.2. Experimental Comparison on ChaLearn Gesture Dataset

The ChaLearn Gesture dataset is a large data set for analyzing body language that consists of 23 h of Kinect video, where 27 subjects perform 20 actions. This dataset is very challenging because the body movements of many action classes are very similar. 

Each video in the NTU RGB + D only includes one action, while each video in ChaLearn includes multiple actions. The ChaLearn Gesture dataset is better-suited for this reason to online action recognition applications. The dataset annotates the start and end frames of 11,116 action instances; 75% of videos with labels are used for training and the rest for testing. Considering the large amount of data, one frame is sampled every four frames.

The experimental results are shown in Table 2. The RPILNet performs well at different observation rates. Even if the observation ratio is only 10%, its accuracy is still higher than other methods. 

The FSNet appears to be more sensitive under different scales. These results further prove that the RPILNet is effective for online applications.

### 4.3. Experimental Comparison on G3D Dataset

The G3D dataset contains 20 movements collected using Kinect cameras. There are 209 uncut long videos in the dataset. In this study, we used 104 of them for training and the rest for testing. Again, the RPILNet performed very well. Our model appears to be well-suited to the G3D dataset as well as to the network structure changes operated through the proposed evaluation process. The experimental results are shown in Table 3.

### 4.4. Ablation Experiment

We conducted ablation experiments to determine the effectiveness of each part of the model. The transformer, capsule network, and both of them were eliminated in turn to run comparative experiments. The results are shown in Table 4.

RPILNet-with FC uses a fully connected layer to replace the transformer and capsule network, then uses the fully connected layer to analyze the features extracted by DenseFusion. 

RPILNet-with capsule and FC uses a fully connected layer to replace the transformer, but retains the capsule network so that each layer can retain the space coordinates and attributes of the capsule.

Finally, RPILNet-with transformer uses the transformer to analyze the extracted features, but does not use a capsule network—an individual layer does not include capsules or their attributes.

The accuracy based on the capsule network and transformer is higher than that of the method without the capsule network or transformer. Moreover, the capsule network contains the attributes of the extracted target (e.g., spatial coordinates and angles).

The accuracy of the model decreases slightly when the observation ratio is 100%. This may be attributable to interference between two adjacent actions. 

This experiment also shows that the RPILNet satisfies real-time operation requirements (>30 fps) under the conditions of the RTX2080S graphics card and the Kinect V2 camera.

### 4.5. Comparative Experiment Based on RGB Image

To verify the generalization ability of the model, we adjusted its structure and retrained it. We chose the action recognition model based on RGB images (no more than five years old) as the object of comparison. We compared them on the three datasets UCF101, Hollywood2, and YouTube.

Figure 9 shows the adjusted model. We believe that the background also contains useful information. For example, if there is a dining table, then the target may be eating. After detecting the ROI area, we extracted the background and replaced the depth image in the original model as one of the inputs.

The experimental results are shown in Table 5.

We found that the proposed method achieves similar performance on UCF101 and YouTube datasets with the current latest method after adjustment. Its accuracy on the Hollywood2 dataset surpasses the other state-of-the-art methods we tested.

## 5. Conclusions

The proposed combination of the capsule network and transformer shows favorable training effects in action recognition. The running speed of the proposed algorithm also satisfies the needs for practical application. By analyzing images, the dependence of the traditional model on the random features extracted by CNN is effectively reduced. The transformer analyzes all input features and extracts new features to improve the interpretability and operability of the network over the traditional approach without sacrificing the accuracy of the CNN. The transformer has unique advantages over the LSTM. While analyzing each element through the attention mechanism, other elements in the sequence model are used as a reference for obtaining new features. The CNN module, which is often used to process images as a tool for sequence analysis, resolves the problem that traditional RNN models are not easily parallelized.

The results of this study confirm that the NLP concept can be used as a tool for analyzing images. We were able to exploit the CNN’s parallelism to preprocess an image while remedying its over-sensitivity to angle changes and noise with the transformer and capsule network. This allows the proposed model to read the content of the image rather than simply observing its pixels.

## Figures and Tables

**Figure 1 sensors-21-02217-f001:**
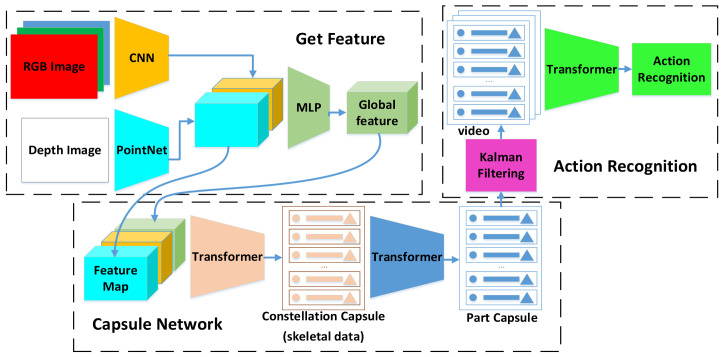
Overview of RPIL model. DenseFusion first extracts features, then constellation capsules representing 25 joint points are obtained via transformer high-level capsules are derived from constellation capsules. Finally, the Kalman filter prevents incorrect capsules from affecting the results.

**Figure 2 sensors-21-02217-f002:**
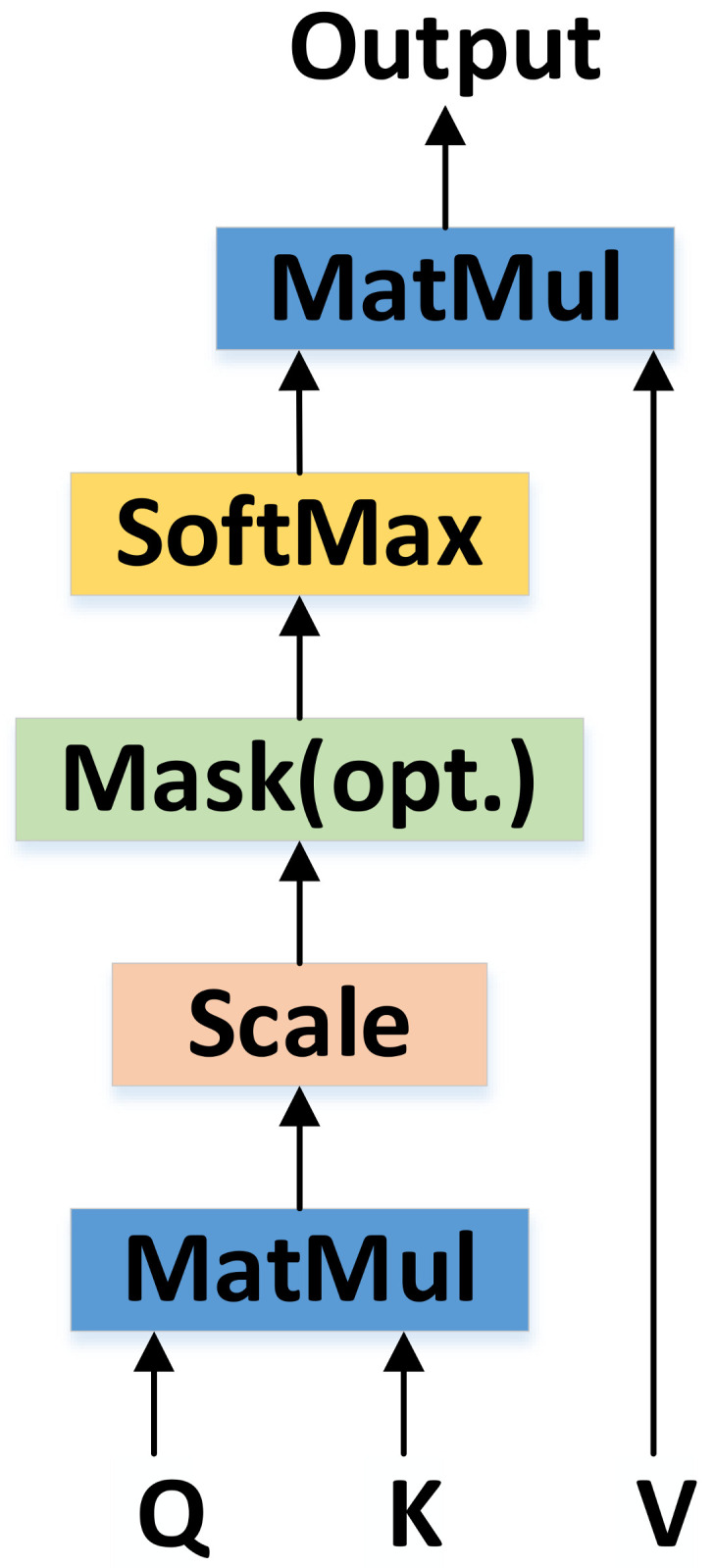
Transformer takes all parts as input and outputs predictions for all targets. Each part is represented by a vector; vectors are influenced by other vectors when they are input to the transformer. The transformer is invariant to permutations of parts.

**Figure 3 sensors-21-02217-f003:**
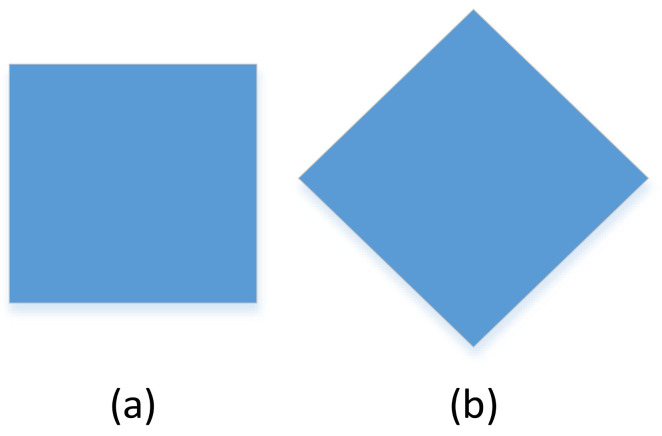
A square can be perceived as a diamond with an angle of 90° or a rectangle with the same length and width by observing changes in its angle. (**a**) A square can be considered a rectangle with equal sides, (**b**) A square can be perceived as a diamond with an angle of 90°.

**Figure 4 sensors-21-02217-f004:**
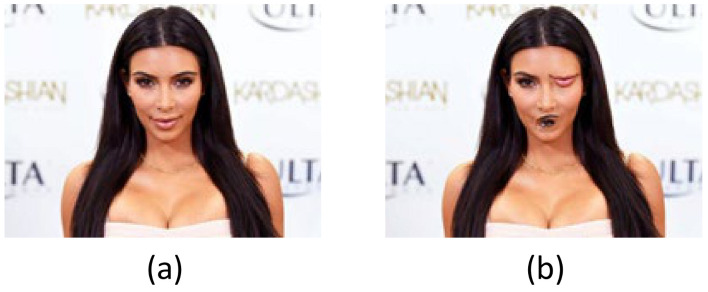
Face recognition depends on the relative position of facial features, not simply whether the features are activated. (**a**) The test result we want, (**b**) CNN may consider some wrong results to be correct results. If it does not consider the coordinates between parts.

**Figure 5 sensors-21-02217-f005:**
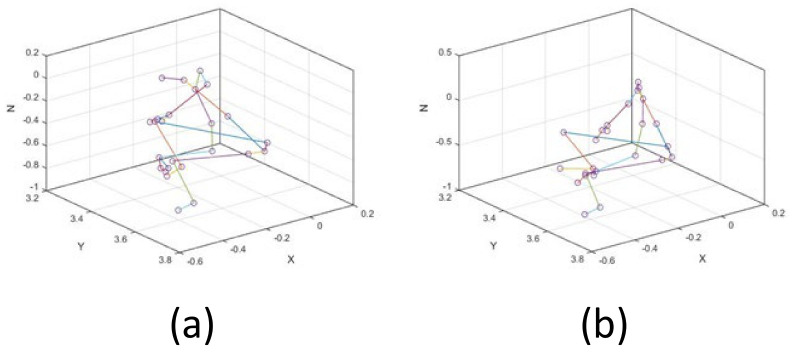
(**a**) One frame of standing posture; (**b**) One frame of sitting posture.

**Figure 6 sensors-21-02217-f006:**
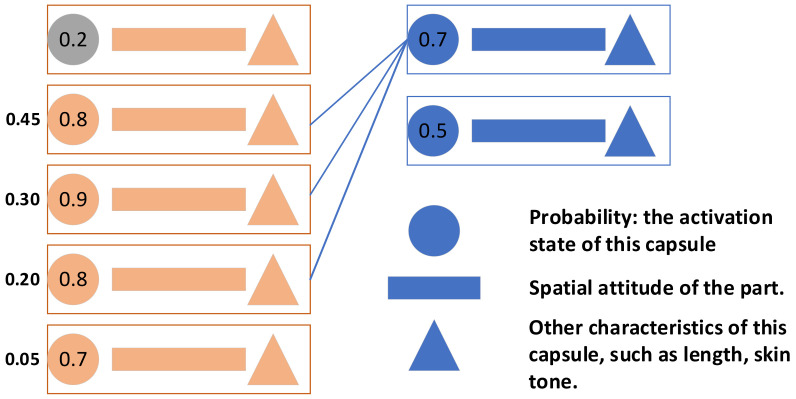
First low-level capsule is not activated; second to fourth low-level capsules contribute to first high-level capsule with cumulative sum of 0.9 > 0.85. Fifth low-level capsule (less contribution) is discarded while second to fourth low-level capsules create activation and characteristic conditions for first high-level capsule. The neural network is retrained to accelerate the algorithm during application.

**Figure 7 sensors-21-02217-f007:**
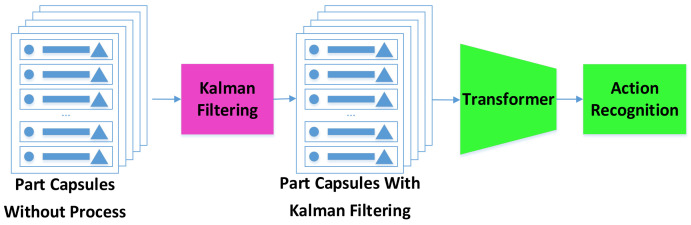
Model checks capsule parameters via Kalman filter to prevent single erroneous capsule in a frame from affecting the final results. The motion trajectory of the capsule is continuous, so the motion range of the capsule at the next moment is set with the coordinates predicted by the Kalman filter as the center. If the coordinates of the capsule are within the range at the next moment, the parameters of the capsule are considered to be correct.

**Figure 8 sensors-21-02217-f008:**
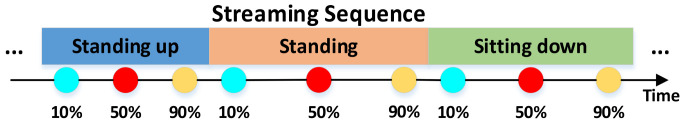
Untrimmed streaming sequence containing multiple action instances. Current ongoing action is to be recognized at each time step when only a portion (e.g., 10%) is utilized.

**Figure 9 sensors-21-02217-f009:**
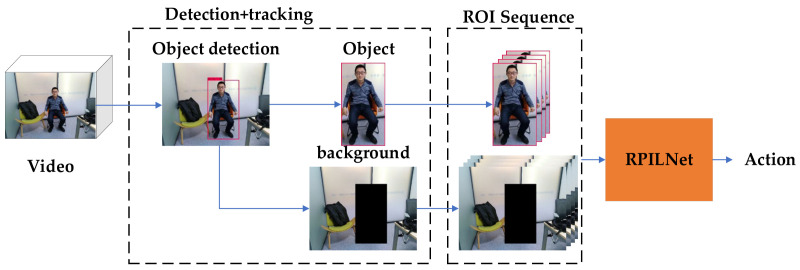
Modified RPILNet works on RGB dataset; object background rather than depth image is input to proposed model.

**Table 1 sensors-21-02217-t001:** Various models on OAD dataset.

Observation Ratio	10%	50%	90%
ST-LSTM [4]	60.00%	75.30%	77.50%
JCR-RNN [5]	62.00%	77.30%	78.80%
Attention Net [6]	59.00%	75.80%	78.30%
FSNet (15) [19]	58.50%	75.40%	75.90%
FSNet (31)	63.30%	75.20%	76.20%
FSNet (63)	62.20%	77.10%	78.90%
FSNet (127)	63.60%	76.30%	78.90%
FSNet (255)	57.20%	70.30%	71.20%
FSNet-MultiNet [19]	62.60%	79.10%	81.60%
SSNet [19]	65.80%	81.30%	82.80%
RPILNet	67.20%	82.10%	84.30%

**Table 2 sensors-21-02217-t002:** Various models on ChaLearn Gesture dataset.

Observation Ratio	10%	50%	90%
ST-LSTM [4]	15.80%	51.30%	65.10%
JCR-RNN [5]	15.60%	51.60%	64.70%
Attention Net [6]	16.80%	52.10%	65.30%
FSNet (15) [19]	16.60%	50.80%	62.00%
FSNet (31)	16.90%	53.20%	64.40%
FSNet (63)	15.80%	49.80%	60.80%
FSNet (127)	14.80%	46.40%	56.40%
FSNet (255)	14.50%	45.70%	55.40%
FSNet-MultiNet [19]	17.50%	54.10%	65.90%
SSNet [19]	19.50%	56.20%	69.10%
RPILNet	19.90%	58.80%	72.20%

**Table 3 sensors-21-02217-t003:** Various models on G3D Dataset.

Observation Ratio	10%	50%	90%
ST-LSTM [4]	67.30%	75.60%	76.80%
JCR-RNN [5]	70.00%	79.10%	81.90%
Attention Net [6]	67.40%	76.90%	79.30%
SSNet [19]	72.00%	81.20%	83.70%
RPILNet	76.20%	84.10%	86.50%

**Table 4 sensors-21-02217-t004:** Ablation experiment based on CHALEARN GESTURE DATASET.

Observation Ratio	10%	50%	90%
RPILNet-with FC	14.90%	50.50%	65.10%
RPILNet-with capsule and FC	18.20%	56.20%	70.20%
RPILNet-with transformer	17.70%	53.90%	67.60%
RPILNet	19.90%	58.80%	72.20%

**Table 5 sensors-21-02217-t005:** Various models on UFC101, Hollywood2 and Youtube.

Method	UCF101	Hollywood2	Youtube
Wang et al. [58]	89.7%	70.6%	78.2%
Yang et al. [59]	92.6%	-	-
Rashwan et al. [60]	78.43%	87.94%	-
Avola et al. [61]	96.2%	-	-
Sharif et al. [62]	-	-	98.2%
Asghari et al. [63]	98.4%	-	-
Almaadeed et al. [64]	98.66%	91.32%	97.65%
RPILNet	97.51%	93.14%	96.65%

## Data Availability

The data in Table 1, Table 2, Table 3, Table 4 and Table 5 is availability.

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
