# Peer review of "“Reading Pictures Instead of Looking”: RGB-D Image-Based Action Recognition via Capsule Network and Kalman Filter"

_sensors, 2021, doi:10.3390/s21062217_

Round 1

Reviewer 1 Report

This paper has proposed an action recognition framework based on capsule network and Kalman filter, which can mitigate the shortcomings of CNN and improve the computational efficiency. The comments of this paper are as follows:

  1. Overall, this article is well organized and its presentation is clear.
  2. In the section of ‘Introduction’, there has some mistakes with the font setting of the first paragraph. Please edit carefully before submission.
  3. Three different benchmark datasets have been chosen to evaluate the performance of the proposed framework, and a set of ablation experiments have been conducted. However, in these tables of experimental results, please indicate the references number of baseline methods. In particular, it is suggested to make the comparison with recent three year state-of-art relevant works.
  4. The limitation of this proposed module is suggested to present.

Author Response

Hello. First of all, thank you very much for your review. Thanks to your suggestions and questions, I can make further revisions to the paper. In response to your suggestions and questions, I have made the following modifications and additions:

  1. Modified and checked the font of section of “Introduce”.
  2. For the comparison experiment of the RGB-D dataset, the method to be compared is supplemented with the reference number in the table.
  3. Complementary explanations are given in section 4.4 for the comparison objects of the ablation experiment.
  4. The model was adjusted and retrained, and compared with the RGB image-based action recognition model in the past five years. The experiment is supplemented in section 4.5. Through experiments, it is found that the method proposed in this paper can achieve similar performance with the latest methods on UCF101 and YouTube datasets. And the accuracy of the Hollywood2 dataset surpasses the current state-of-the-art methods.
  5. Limitations of the method: The model improves the accuracy by using the transformer, but makes the model more complex. The running speed of the model is slower than that of the model using the fully connected layer. But through the pruning process of section 3.3, the speed of the model can be improved to a certain extent under the premise of ensuring accuracy. This part is supplemented in the overview of Section 3.
  6. The modified place has been marked in yellow.

Finally, I want to express my gratitude for your help. Your suggestions help us to make the experiment and paper more complete.

Thank you for your review and guidance.

Best regard.

Botong Zhao

Reviewer 2 Report

1. What is the difference between action recognition (in title) and posture detection (in Conclusion) in this paper? 
2. It would be better to place Figure 1 in Sec. 3. Explain it and refer to it in the text body.
3. " This method resolves the convolutional neural network's over-sensitivity to rotation and scaling, and increases the interpretability of the model as per the spatial coordinates in graphics." From the whole manuscript, I cannot find experimental results or analysis to support this point.
4. Sec. 3 lacks the overall description and analysis of the proposed method.
5. In Abstract "The RPILNet also satisfies real-time operation requirements (>30 fps)." 
    L375-376, 4.4. Ablation experiment, "This experiment also shows that the RPILNet satisfies real-time operation requirements (>30 fps)."
   Which experiment? What are the results to support ">30 fps"?
6. The hardware and software environments are not given in this paper. Without this information, the frame ratio is meaningless.
7. Why Kalman Filter is used? The authors must explain it.
8. How to understand the word "reading" in the title? Does NLP play a role in the proposed model?

Author Response

Hello. First of all, thank you very much for your review. Thanks to your suggestions and questions, I can make further revisions to the paper. In response to your suggestions and questions, I have made the following modifications and additions:

1. The "posture detection" in the paper I want to express is the title of "action recognition". There is no difference between them. The "posture detection" in the article has been modified to "action recognition".

2. Figure 1 is placed in section 3 and explained in the text body.

3. For the experimental description of capsule network overcoming angle sensitivity, I added it in section 4.4. Through comparison, it can be found that the accuracy of the method using the capsule network is higher than that of the method without using the capsule network. Moreover, the capsule network contains the attributes of the extracted objects, such as spatial coordinates and angles. This method allows readers to more intuitively understand the relationship between two adjacent neural layers. And this is why I chose RGB-D-based action recognition as the application direction of the paper.

Although the capsule network can extract parts and combine them in other data sets, such as coco, mnist and other datasets. But we believe that the joints of the human body can better help readers understand "the effectiveness of the spatial coordinate system for neural network applications."

For example, the Hough transform will not fail to recognize a straight line after it is rotated. One of the core ideas that the capsule network wants to express is that "neural networks can also use spatial coordinates to process information like imaging."

4. In the overview part of section 3, the overall plan is described and analyzed.

5. The model meets the real-time operating conditions are supplemented at the end of section 4.4.

6. The program runs under the RTX2080S graphics card using the Kinect2 camera, and it is concluded that it meets the real-time requirements during the test.

7. The purpose of using Kalman filtering is to check the predicted results. We believe that the action of the target is continuous, and the use of Kalman filtering can avoid the erroneous prediction at a certain moment from having an uncertain effect on the result of the entire sequence. Our goal is to use the predicted results to constrain the range of motion of the target at the next moment. These instructions are placed in the first paragraph of section 3.4.

8.We use the word "reading" because we believe that the process of analyzing a picture is not to look at every pixel of the picture, but to analyze the information contained in different positions of the picture like reading an article. We think this process is very similar to the sequence model. So we choose to use transformer to analyze the capsule network. For these descriptions, we will make supplementary descriptions in the overview of section 3.

9. The modified place has been marked in yellow.

Finally, I want to express my gratitude for your help. Your suggestions help us to make the experiment and paper more complete.

Thank you for your review and guidance.

Best regard.

Botong Zhao

Reviewer 3 Report

This paper presented an action recognition algorithm based on the capsule network and  Kalman filter called “Reading Pictures Instead of Looking” (RPIL). Authors used Kalman filter and capsule network and Bert to analyze the parameters.

  • Abstract has been well written with the performance.
  • In main body, authors used the natural language processing (NLP) concept. But this was not clear to understand. 
  • In experimental results, the performance were compared with a little old methods (2013~2016).  It would be better to compare the performance with more recent methods in 2019, 2020. 
  • There are many works for classification as: 

        -A Novel Approach for Robust Multi Human Action Recognition and Summarization based on 3D Convolutional Neural Networks,  https://arxiv.org/pdf/1907.11272.pdf

         - A Comprehensive Survey of Vision-Based Human
Action Recognition Methods, Sensors, 2019 

         - Deep Joint Spatiotemporal Network (DJSTN) for Efficient Facial Expression Recognition, Sensors, vol. 2020, March 2020

Author Response

Hello. First of all, thank you very much for your review. Thanks to your suggestions and questions, I can make further revisions to the paper. In response to your suggestions and questions, I have made the following modifications and additions:

1. We use the concept of "NLP" because we believe that the process of analyzing a picture is not to look at every pixel of the picture, but to analyze the information contained in different positions of the picture like reading an article. We think this process is very similar to the sequence model. So we choose to use transformer to analyze the capsule network. For these descriptions, we will make supplementary descriptions in the overview of section 3.

2. For the experimental description of capsule network overcoming angle sensitivity, I added it in section 4.4. Through comparison, it can be found that the accuracy of the method using the capsule network is higher than that of the method without using the capsule network. Moreover, the capsule network contains the attributes of the extracted objects, such as spatial coordinates and angles. This method allows readers to more intuitively understand the relationship between two adjacent neural layers. And this is why I chose RGB-D-based action recognition as the application direction of the paper.

3. The model was adjusted and retrained, and compared with the RGB image-based action recognition model in the past five years. The experiment is supplemented in section 4.5. Through experiments, it is found that the method proposed in this paper can achieve similar performance with the latest methods on UCF101 and YouTube datasets. And the accuracy of the Hollywood2 dataset surpasses the current state-of-the-art methods.

4. The modified place has been marked in yellow.

Finally, I want to express my gratitude for your help. Your suggestions help us to make the experiment and paper more complete.

Thank you for your review and guidance.

Best regard.

Botong Zhao

Round 2

Reviewer 2 Report

My comments in the first round review are well answered. I think this paper can be accepted now.

The authors need to carefully check the information of the references, such as the authors' first names and last names. 

Reviewer 3 Report

As my check, this paper has been well revised based on the comments.  I have no further comments.